# Enhancing the TiO₂-Ag Photocatalytic Efficiency by Acetone in the Dye Removal from Wastewater

**Catalina Nutescu Duduman [1], Consuelo Gómez de Castro [2], Gabriela Antoaneta Apostolescu [1], Gabriela Ciobanu [1], Doina Lutic [3,*], Lidia Favier [4] and Maria Harja [1,*]**

[1] Department of Chemical Engineering, Faculty of Chemical Engineering and Environmental Protection, Gheorghe Asachi Technical University of Iasi, 73, Prof.dr.doc. D. Mangeron Blvd., 700050 Iasi, Romania

[2] Department of Materials and Chemical Engineering, Faculty of Chemical, Complutense University of Madrid, Av. Séneca, 2, 28040 Madrid, Spain

[3] Faculty of Chemistry, Alexandru Ioan Cuza University of Iasi, Blvd. Carol I No 11, 700506 Iasi, Romania

[4] Univ Rennes, Ecole Nationale Supérieure de Chimie de Rennes, CNRS, ISCR—UMR6226, F-35000 Rennes, France

\* Correspondence: doilub@uaic.ro (D.L.); mharja@tuiasi.ro (M.H.)

**Abstract:** TiO₂ nanoparticles synthesized by the sol-gel method and doped with Ag were characterized by SEM, EDAX, FTIR, BET, XRD and TEM, then tested in the photocatalytic degradation of methylene blue (MB) under UV irradiation. The experimental results indicate that the average size of the raw particles was 10 nm, and their size was increased by calcination. The photocatalytic degradation of MB on nanostructured TiO₂-Ag shows a high degradation efficiency upon the addition of a photosensitizer. A parametric study of the process was performed and has revealed the optimal value of the photocatalyst dose (0.3 g L⁻¹) at a MB concentration of 4 ppm. Afterwards, the effect of acetone as a photosensitizer was studied. A MB degradation mechanism was proposed to explain the synergy between the TiO₂ and the silver nanoparticles in the degradation performance. Under the optimal experimental conditions, at photosensitizer doses of 0.1 and 0.2%, yields of 92.38% and 97.25% MB degradation were achieved, respectively. Kinetic models showed that, at 0.1% acetone concentration, the data fit the pseudo-first-order model, while at 0.2% acetone, the photodegradation mechanism fits a second-order model. The values of the apparent rate constants indicate that the reaction rate increased between 24 and 40 times in the presence of acetone on TiO₂ and TiO₂-Ag. The addition of acetone modified the photodegradation mechanism and the Ag-doped samples became more active. The results of recycling tests using calcined TiO2-Ag material clearly show that the material was highly photocatalytically stable for the MB degradation. According to experimental results, the dye degradation decreased from 97.25% to 92.39% after four consecutive cycles. This simple approach could be applied for the advanced cleaning of wastewater contaminated with dyes, in the perspective of its reuse.

**Keywords:** TiO₂-Ag nanoparticles; methylene blue degradation; acetone photosensitizer; kinetic modeling

## 1. Introduction

The contaminants found in wastewater, including heavy metal ions, organic compounds, microorganisms, etc., have raised serious concerns owing to their significant negative effects on the environment and the economy [1–4].

Numerous synthetic and very stable dyes are extensively used in the textile, carpets and leather industries, paper production, automotive and military industries, food conditioning, cosmetics, toys, etc. [5,6]. According to several estimations, over $7 \times 10^5$ tons of dyes are produced annually worldwide. Unavoidably, during the production and the normal use of colored objects, significant amounts of dyes are detached from their surface or mass, and sooner or later reach water effluents [7], causing considerable environmental pollution, and serious risk of eutrophication or the generation of toxic metabolites by their

oxidation and hydrolysis [8,9]. Additionally, dyes are known for their toxicity for humans and aquatic system, and their presence in contaminated effluents contributes to reductions in the light penetration path [8].

In recent years, water reuse has been regarded as an outstanding strategy, provided that a proper cleaning treatment is applied [10–12]. Numerous approaches have been conceived for the removal of synthetic dyes from wastewaters, mainly based on extraction with selective solvents, adsorption, and chemical treatments, applied prior to the biological step of wastewater cleaning. However, these procedures raise new environment problems, since they use chemical reagents (for precipitation or coagulation–flocculation), and generate solid waste, transferring the pollution problem from wastewater to a solid phase. An advantageous alternative would be the irreversible destruction of troublesome chemicals, and photocatalytic degradation delivers high benefits in this respect [10,13,14]. An important advantage of photocatalysis is that the process allows the complete mineralization of persistent organic pollutants (POPs) at high rates, even at ambient temperatures, with minimal energy consumption and little toxic products generation, by using mainly atmospheric oxygen as an oxidizer. This strategy allows the development of basic, low-cost wastewater recycling systems [10] in a circular economy.

Owing to their high potential applicability in wastewater treatment in a relatively cheap and green manner, the photocatalytic materials are now being intensively investigated [15–17]. Titanium dioxide ($TiO_2$) is the most used solid for photocatalytic reactions, due to its high photocatalytic efficiency, biological and chemical inertness, low toxicity and easy availability at an acceptable price [18]. Titanium dioxide mainly exists in three crystallographic phases: rutile, anatase and brookite. For photocatalytic purposes, anatase and rutile are better than brookite; their band gap values are of 3.02 and 3.2 eV, respectively [19,20].

The photocatalytic activity of $TiO_2$ can be enhanced by several methods: doping with noble metals of other semiconductive oxides [21], surface modification [19], shape tailoring, etc. An ambitious target for researchers in photocatalysis is to formulate solids activatable by solar light; this contains about 3% UV and 97% visible radiation, and is entirely green, easily available and extremely cheap. Metal doping and heterojunction formation on the $Me/TiO_2$ composites exhibit band gap values slightly below 3 eV, allowing the promotion of the electrons from the valence to the conduction bands by capturing light from the solar spectrum [17]. The noble metals (Pt, Ag, Au, Pd) and several transition metals (Mo and Ru) have been the most commonly investigated in this respect [21]. Between the noble metals, silver is particularly attractive due to its remarkable catalytic activity [22], and it is economically more affordable. The Ag nanoparticles loaded from salts on oxide supports lead to the formation of $Ag^0/Ag+$ couples on the surfaces after the partial reduction of Ag ions, enhancing the photocatalytic activity [23–26]. $TiO_2$-Ag is well-known for its antibacterial properties, inactivation of bacteria, viruses and molds, etc. [27], but also for its photocatalytic activity [28]. The $TiO_2$ synthesis methods and conditions play a decisive role in nanostructure development. Among numerous chemical synthesis methods applied for obtaining solids, the sol-gel approach is relatively simple, low-cost, easily operated, and allows for the good control of experimental conditions; therefore, expected shapes, uniform sizes and high purities are obtained [19]. $TiO_2$ containing Ag nanoparticles with high homogeneity and controlled particle sizes is easily obtained by the sol-gel method [28].

According to the literature, the photodegradation of synthetic dyes in aqueous solutions could be much enhanced by using photosensiting species. Among these, acetone and hydrogen peroxide are mentioned as photosensitizers and/or ●OH sinks [29,30].

The aim of this study was to synthesize $TiO_2$- and Ag-doped $TiO_2$ samples, by the sol-gel method, from titanium tetraisopropoxide and silver nitrate, characterize the samples and evaluate their photocatalytic performance in the photocatalytic degradation of aqueous solutions of methylene blue (MB), selected as the model molecule, under UV irradiation. The beneficial effect of acetone added as a photosensitizer was validated, and the photodegradation reaction kinetics in various systems were investigated.

## 2. Materials and Methods

**Reagents and preparation.** The sol-gel method was employed to obtain $TiO_2$ nanoparticles, using titanium tetraisopropoxide (TTIP—$C_{12}H_{28}O_4Ti$) Sigma-Aldrich as the precursor [28]. The Ag doping was performed using silver nitrate ($AgNO_3$) (Chemical Company, Iasi, Romania) the as Ag source, and hydrazine ($N_2H_4$) (Spectrum) as the reducing agent. Nitric acid ($HNO_3$) (Sigma-Aldrich, St. Louis, MO, USA) 65% served as the hydrolysis condensation-regulation agent and ammonia solution (0.1 M) was used as the neutralizer.

For the preparation, TTIP was diluted in half of the ethanol (98%) (Chemical Company), then nitric acid was added dropwise. The formation of Titania nanoparticles was achieved by bringing the pH to a value between 10 and 11 by adding ammonia. The doping procedure with Ag consisted of stirring the $TiO_2$ suspension prepared as described before for 2 h, then adding the $AgNO_3$ aqueous solution (1 M). Hydrazine (diluted in 5 mL ethanol) was added after 1 h and stirring continued another 1 h. The resulted product was filtered and dried at 110 °C. Part of it was calcined for 2 h at 650 °C (S3) [28,31]. The recipes for the preparations are presented in Table 1.

**Table 1.** Experimental conditions for obtaining the photocatalysts.

| Sample | TTIP (mL) | Ethanol, (mL) | HNO$_3$ (mL) | NH$_3$ (mL) | AgNO$_3$, (g) | Hydrazine, (g) | Calcination |
|--------|-----------|---------------|--------------|-------------|---------------|----------------|-------------|
| S1 | 5.8 | 10 | 0.50 | 100 | - | 1.33 | - |
| S2 | 5.8 | 10 | 0.50 | 100 | 0.418 | 1.33 | |
| S3 | 5.8 | 10 | 0.50 | 100 | 0.418 | 1.33 | 650 °C/2 h |

**Sample characterization.** The morphological characterization and elementary chemical composition were examined on JEOL 6400 with an Oxford Link EDAX microanalyser (JEOL JSM-7100, Jeol Ltd., Tokyo, Japan). The TEM images were obtained on a JEOL JEM-2100 machine (JEOL JEM-2100, Jeol Ltd., Tokyo, Japan). The XRD patterns were collected on a X'Pert PDP3040 Philips apparatus (X'Pert PDP3040 Philips, Austin, TX, USA) and X'Pert HighScore Plus PANalytical dedicated software was used for the interpretation of the diffraction patterns. The Nova 2200 apparatus allowed the determination of the specific surface area using the BET equation (Quantachrome Instruments, Graz, Austria).

**Photodegradation experiments.** Methylene blue (MB) (Merck) is a cationic dye with the molecular formula $C_{16}H_{18}ClN_3S$ (molar mass 319.85 g/mol) and the absorption characteristic peak $\lambda_{max}$ of 664 nm. The MB concentrations were measured by spectrophotometric measurements on a UV–Vis 1800 Shimadzu spectrophotometer (UV-VIS 1800 Shimadzu, Duisburg, Germany). In the photocatalytic reaction, the irradiation source was a UV-B lamp with Hg (18 W) (OSRAM GmbH, Garching, Germany), emitting an incident radiation of 2.1 W/m$^2$ intensity (calculated from the distance between the samples and the light source). The experiments were performed in triplicate in a 500 mL reactor, at ambient temperature (25 °C). The photocatalyst (S1, S2 or S3) and 250 mL of 4 ppm MB solution were placed into the reactor and stirred for 25 min in the dark, to reach the adsorption equilibrium. The reactions were performed without adjusting the native solution pH (6.5). At the appropriate time values, samples were taken, centrifuged to separate the solid, and then the absorbance of the supernatant was read three times, noting the average value.

The degradation extent of MB was calculated using the relation:

$$R\ (\%) = 100\ (C_0 - C_t)\ /\ C_0 = 100\ (A_0 - A_t)/A_0,\ \% \tag{1}$$

where R (%) is the dye degradation yield, $C_0$ and $C_t$ are the initial and t-time concentrations of MB, and $A_0$ and $A_t$ are the absorbance values at the same time values.

The mineralization of MB was expressed as the percentage of total organic carbon (TOC) reduction, calculated from the difference between the initial and final TOC content measured for the aqueous dye solution:

$$\text{Mineralization (\%)} = \frac{\text{TOC}_{\text{initial}} - \text{TOC}_{\text{final}}}{\text{TOC}_{\text{initial}}} \cdot 100 \tag{2}$$

For the total organic carbon measurement, TOC-L Shimadzu equipment (TOC-L Shimadzu, Milano, Italy) was used.

## 3. Results and Discussion

### 3.1. Characterization of the Photocatalysts

The crystalline nature of the TiO$_2$–Ag nanoparticles was determined by XRD; the patterns are shown in Figure 1. The peak locations and relative intensities for TiO$_2$ were assigned using the JCPDS database.

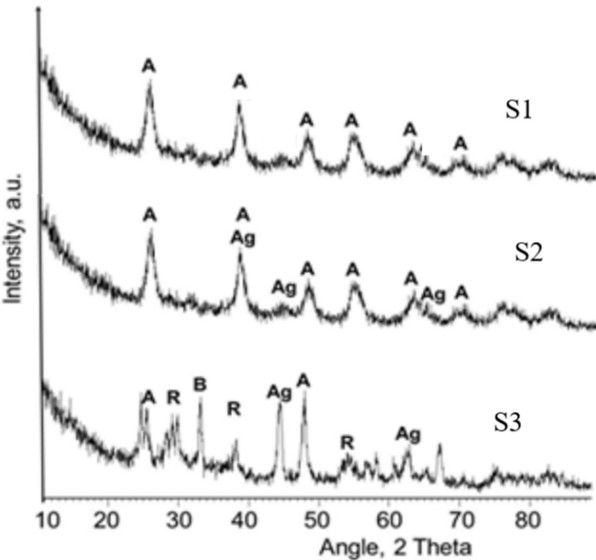

**Figure 1.** XRD patterns of the samples. A—anatase, R—rutile, B—brookite and Ag—silver.

The powder XRD results of sample S1 indicate the formation of anatase crystalline phase, mixed with amorphous phase (pattern not shown). The XRD pattern of S2 reveals a partially crystalline solid, as shown by the presence of the main characteristic peaks of anatase from 2θ of 25.3° (101), 38° (004), 48° (200), 54° (105), 62.5° (204) and 69° (116). Although the strongest peak due to silver at 38.1° (111) overlaps with that of anatase, the presence of Ag as a distinctive phase was proven by the peaks from 44.5° and 64.4°, associated with the (200) and (220) planes of metallic silver [32,33].

After the calcination, the XRD pattern indicate a large extent of crystallinity increase, and the peaks of S3 are higher and narrower. Moreover, some of the anatase was transformed to rutile phase, as shown by the maxima from 27° (110), 36° (101) and 55° (211). The peaks due to silver almost disappear, suggesting that the calcination induced the fine dispersion of Ag on the surface of TiO$_2$, and the corresponding nanoparticles are not visible as bulk phase.

The SEM microscopy images of the S1–S3 samples show the surface morphology (Figure 2). The Titania sample (S1) consisted of polyhedrical grains with relatively uniform sizes of 1–1.5 μm, with lots of sharp edges and irregularities. The influence of doping Ag in S2 had a significant influence on the solid morphology: most grains have sizes of less than 1 μm, giving an image of a rough surface; a rare polyhedral grain of about 10 μm could be detected. Upon calcination (S3), the particles turn to bigger grains of 2–3 μm, with irregular shapes and many edges, and slightly agglomerated.

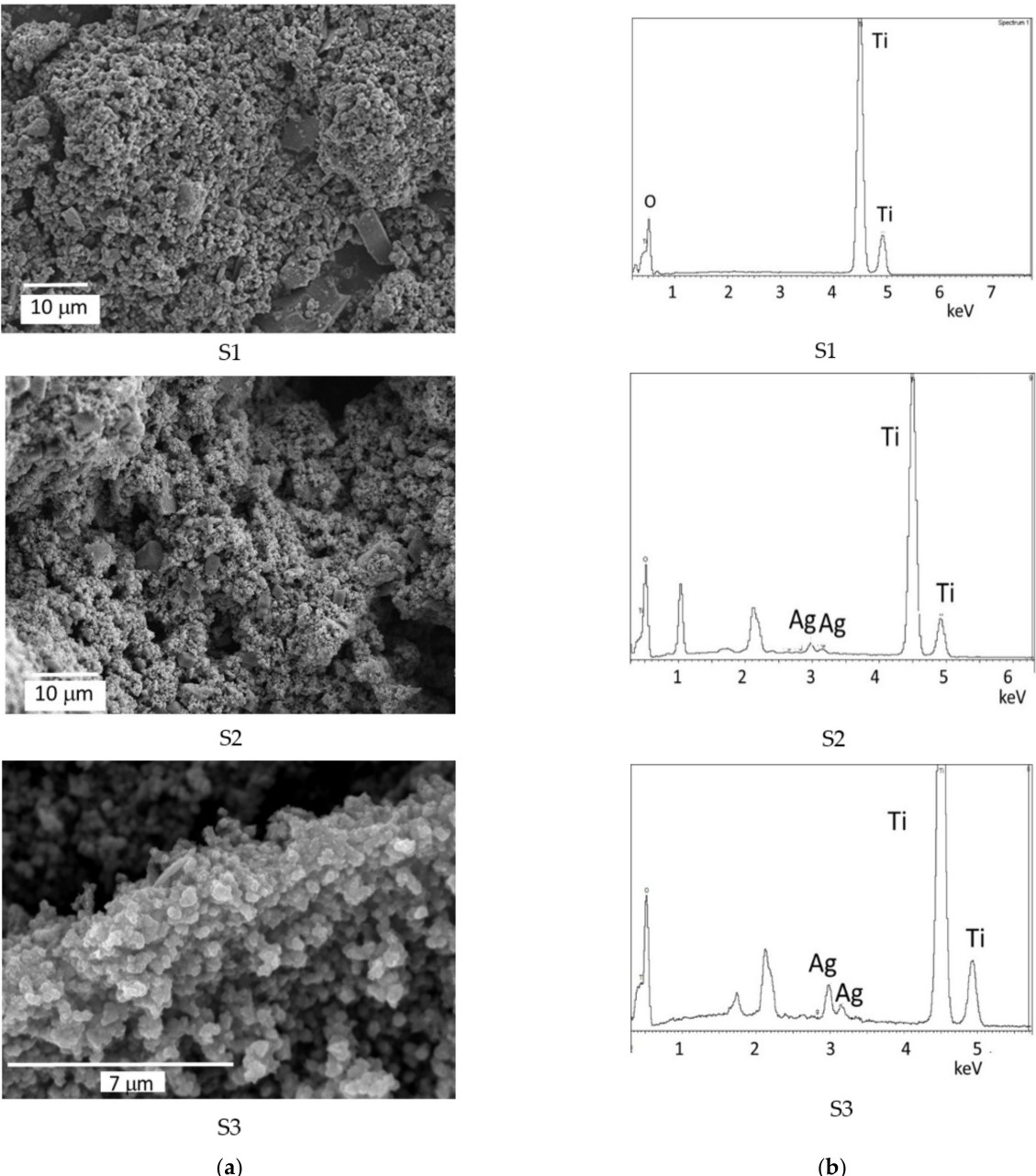

**Figure 2.** (**a**) SEM micrographs and (**b**) EDAX pattern of the S1–S3 samples.

The EDAX analysis proves the presence of Ag in samples S2 and S3, while pure titania was found in S1. It is interesting to note that calcination does not modify the Ag content on the TiO$_2$ surface, despite the different Ag particle sizes shown by XRD.

The fine dispersion of the Ag nanoparticles on titania after the calcination was highlighted by TEM analysis (Figure 3). The Ag nanoparticles are present as small crystalline grains (dark dots) seldom agglomerated, with sizes less than 20 nm. The Ag particle dispersion was not lost during the calcination procedure. The lighted dots and circles in the SAED pictures (Figure 3) indicate that, in both S2 and S3 samples, Ag is in the crystalline state.

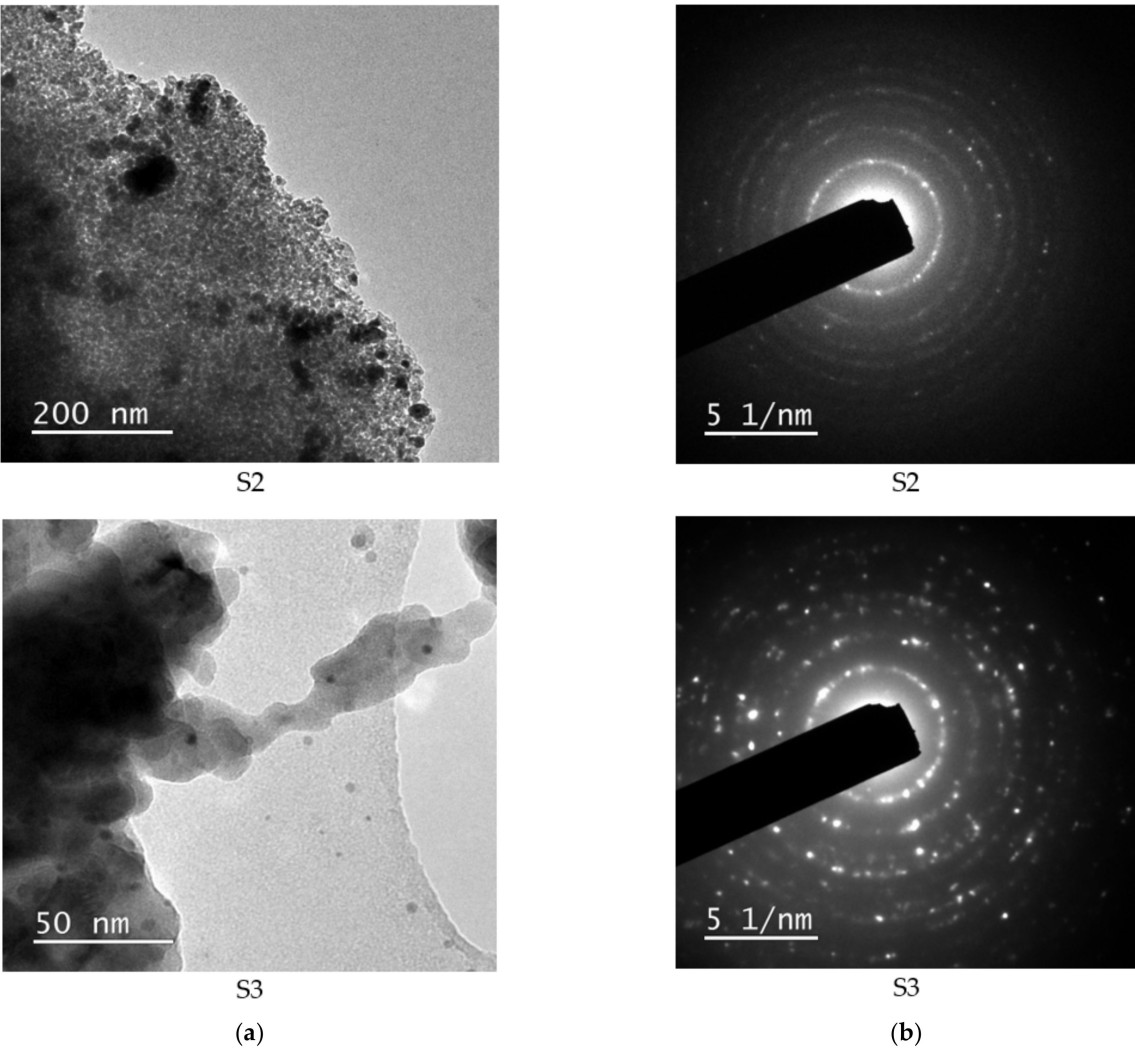

**Figure 3.** (**a**) TEM micrographs and (**b**) SAED images of samples with S2 and S3.

The FTIR spectra of the samples are shown in Figure 4. The broad bands situated between 3500 and 3000 cm$^{-1}$ are due to the stretching vibrations of the hydroxyl groups, while the peak around 1620 cm$^{-1}$ is ascribed to hydroxyl groups bending. The band at 1384 cm$^{-1}$ is assigned to the organic residual compounds embedded in the titania, from the ethanol used in the synthesis and from the isopropanol formed during the hydrolysis step. The large band between 800 and 400 cm$^{-1}$ contains many small peaks, and can be attributed to the stretching vibrations corresponding to the Ti–O or O–Ti–O bondings [34]. In the spectra of S2 and S3 samples, important changes appear. The bands observed between 3500 and 3000 cm$^{-1}$ and 1620 cm$^{-1}$ have higher intensities, indicating the enrichment of the structure in surface hydroxyl groups, as does the band at 1384 cm$^{-1}$, highlighting an increase in the number of sites where organic matter from the two alcohols, as well as from hydrazine, remains bonded. The region between 800 and 400 cm$^{-1}$ was significantly changed due to the Ag nanoparticles present. According to Ahmed et al. [35], the bands at 645 and 725 cm$^{-1}$ are due to Ag nanoparticles.

The surface areas of the samples were, respectively, 20.35 m$^2$ g$^{-1}$ for S1, 25.37 m$^2$ g$^{-1}$ for S2 and 21.69 m$^2$ g$^{-1}$ for S3.

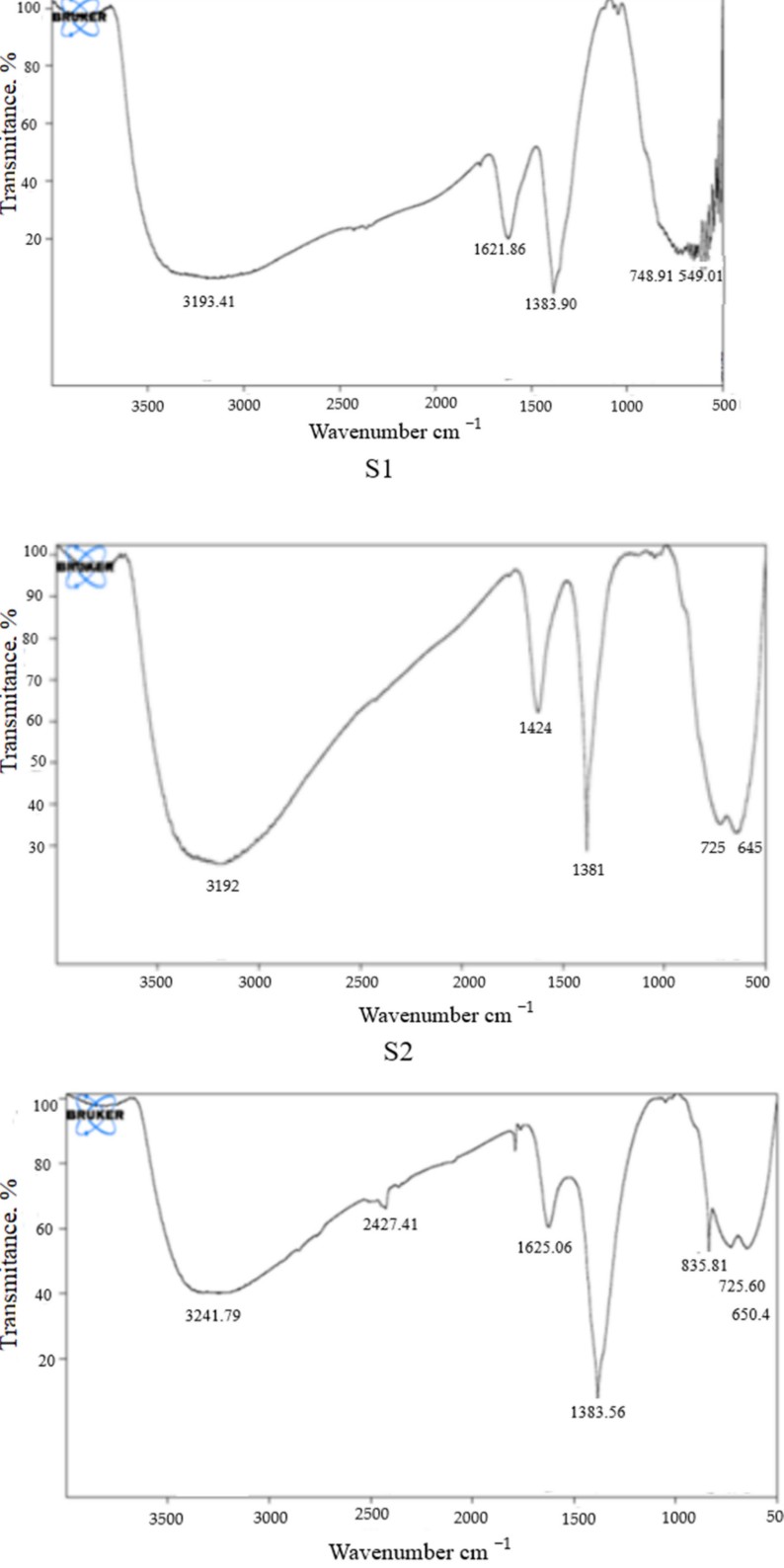

**Figure 4.** FTIR spectra of the samples.

### 3.2. MB Photocatalytic Degradation

The photocatalytic performance of the synthesized solids was evaluated by following the photocatalytic decolorization of MB aqueous solution under UV light irradiation. The UV–VIS spectrum of MB displays a characteristic absorption maximum at λ = 664 nm, which was used to calculate the dye concentration by UV–VIS spectroscopy (Figure 5).

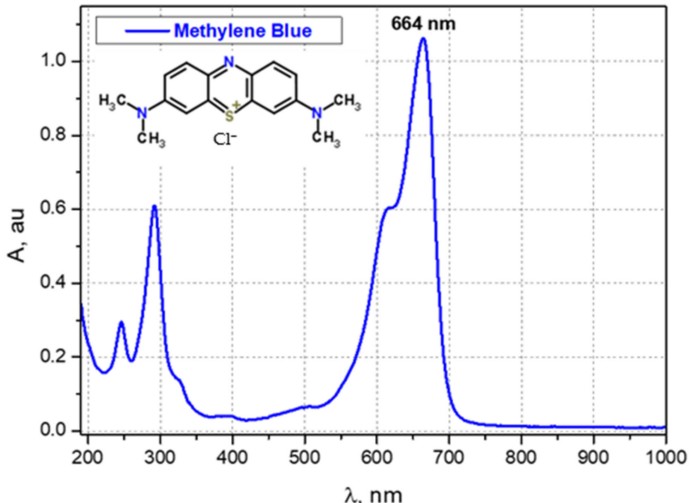

**Figure 5.** Absorbance spectrum of the MB solution (16 ppm).

The effect of the catalyst dosage on the MB degradation was investigated using a dye solution of 4 ppm and photocatalyst doses of 0.1, 0.2, 0.3 and 0.4 g L$^{-1}$, with a reaction run time of 60 min (results not shown). The decolorization yield increased up to 0.3 g L$^{-1}$ catalyst load, then remained constant.

The results of the first series of photocatalytic experiments are displayed in Figure 6, showing MB absorbance evolution with the time of initial solutions (4 ppm) exposed to UV irradiation only (blank), and in the presence of 0.3 g L$^{-1}$ photocatalyst (S1, S2 and S3) under UV irradiation.

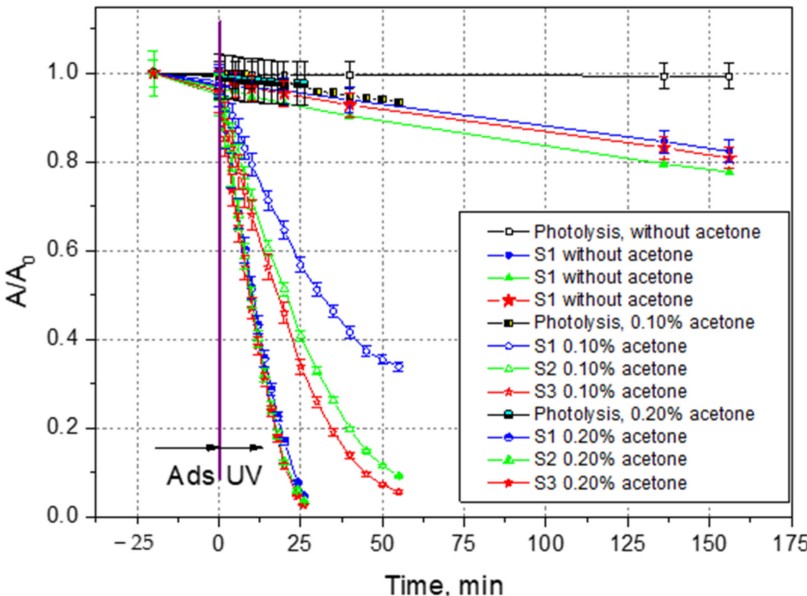

**Figure 6.** MB removal from the solution by photolysis (blank experiment), photocatalysis in the presence of S1, S2 and S3, and effect of acetone addition in the reaction system (4 ppm MB, 0.03 g L$^{-1}$ photocatalyst).

The results illustrate the good stability of MB against UV exposure: only 2% of the initial amount of dye was degraded after 150 min by photolysis; this result is in line with data from the literature [13]. For all three samples, the dye uptake by adsorption was quite low (around 3% of the dye). The photocatalytic performance, however, was not high enough to be worth continuing more detailed parametric studies in these systems, since the decolorizing yields were only 16.9% on S1, 21.7% on S2 and 19.3% on S3, respectively. Although the results are moderate, we see a certain enhancement of the photocatalytic performance in the samples doped with Ag.

The blank experiment run using the photosensitizer and irradiation without photocatalyst revealed that less than 5% of MB was decolorized after one hour under UV exposure. It is also worth observing that adsorption equilibrium was reached during the first 25 min of contact between the dye solution and the solid.

Small amounts of acetone addition in the reaction mass brought a revolutionary improvement in the photocatalytic MB decomposition. Its influence in the three systems containing S1, S2 and S3 photocatalysts was quite different. At a ratio of 0.1% acetone in the reaction mixture, the MB decolorization reached around 52% on S1, 60% on S2 and 64% on S3 after only 25 min of reaction. The differences between the samples S1 and S2, or S3 (containing Ag nanoparticles), became even more visible after 50 min of irradiation, when the color loss was 65% on S1 and over 90% on samples S2 and S3. The calcination does not significantly change the activity of Ag-doped sample. When the acetone ratio increased to 0.2%, the decolorization was very fast: in only 25 min, over 95% of the dye was decomposed.

The participation of the photosensitizer in the photocatalytic reaction was of the utmost importance for MB molecule breakage (loss of color). The essential role of acetone in the process can be more easily illustrated by comparing the decolorization yields after 50 min of UV exposure, without and with the sensitizer (Figure 7).

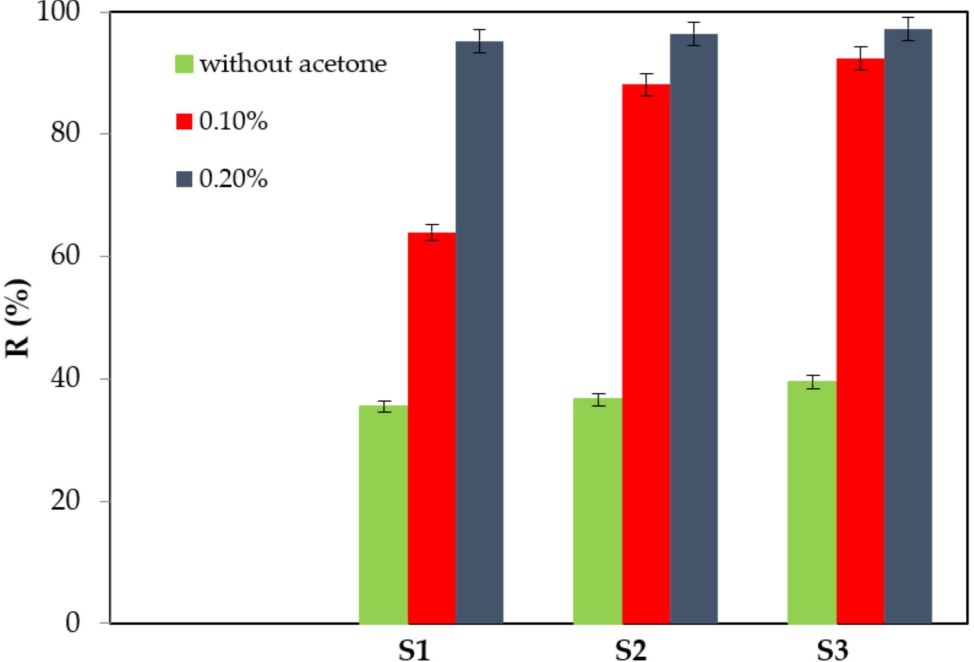

**Figure 7.** Effect of photosensitizer concentration on the MB removal.

The significant improvement in the photocatalytic performances of the samples S2 and S3 seems to be due to a delay in the electron–pair recombination due to the presence of silver; this helps keep the electron promoted on the conduction band of $TiO_2$ on its own conduction band for a while, instead of allowing its transfer back to the valence band of the semiconductor. Thus, the adsorbed $O_2$ has the opportunity to generate superoxide radicals

($O^{2-}$). In the meantime, acetone can split into radicals upon UV irradiation, generating extra OH• radicals in the reaction medium and enhancing the photoactivity. Hence, more radicals are present in the reaction medium, and more dye molecules will be fragmented, thus enhancing the degradation of the target compound (the decolorization values were close, at 95.2%, 96.42% and 97.25%, respectively).

The information gathered in Table 2 reveals the large diversity of materials that can be used as photocatalytic materials for MB degradation.

**Table 2.** Photocatalytic performances of different materials for MB degradation.

| Material | Preparation Procedure | Reaction Conditions | Elimination Efficiency | Ref |
|---|---|---|---|---|
| Au:TiO$_2$ and Cu:TiO$_2$ thin film | Sol-gel | UV and visible light | 80 and 90% | [21] |
| Nanosized SnO$_2$ photocatalysts | Precipitation method chloride dihydrate and isopropyl alcohol | UV Hg lamps 10 ppm MB, 4 mg catalyst | 79% after 180 min | [36] |
| Zinc oxide nanoparticles-decorated graphene oxide (ZnO@GO) | Solvothermal method with zinc oxide nanoparticles and graphene oxide | Neutral solution under UV light irradiation | 98.5% after 15 min | [37] |
| Al, Fe co-doped ZnO (Al–Fe/ZnO) nanorods | Hydrothermal method from zinc, iron and aluminum nitrate | 10 mg L$^{-1}$, continuous magnetic stirring and visible-light Xenon lamp 300 W | 90% MB dye in 75 min | [38] |
| rGO/TiO$_2$ nanocomposite | Ultrasound-assisted | pH value of 13.2 and photocatalyst dosage of 2 g L$^{-1}$ | 91.3% within 30 min | [39] |
| TiO$_2$ and Pd/TiO$_2$ | Sol-gel method (Ti isopropoxide and Pd nitrate) | Methylene blue, methyl orange, 100 W UV lamp | 83.4 and 75.3% after 180 min | [40] |
| TiO2-Ag | Sol-gel method with TTIP and silver nitrate, calcined at 650 °C/2 h | UV-B lamp with Hg (18 W), 0.2% acetone | 97.25% over 20 min | This study |

A series of tests to explain the photocatalytic degradation of MB in the presence of TiO$_2$-Ag and acetone was proposed based on the literature data [41–43].

*Photocatalyst: TiO$_2$*

$$TiO_2 + h\upsilon \ (UV) \rightarrow TiO_2(e_{CB}^- + h\upsilon_B^+) \tag{3}$$

$$TiO_2 \ (h\upsilon_B^+) + H_2O \rightarrow TiO_2 + H^+ + HO^\bullet \tag{4}$$

$$TiO_2 \ (h\upsilon_B^+) + OH^- \rightarrow TiO_2 + H^+ + HO^\bullet \tag{5}$$

$$TiO_2 \ (e_{CB}^-) + O_2 \rightarrow TiO_2 + O_2^{\bullet-} \tag{6}$$

$$O_2^{\bullet-} + H^+ \rightarrow HO_2\bullet \tag{7}$$

$$HO_{2\bullet} + HO_{2\bullet} \rightarrow H_2O_2 + O_2 \tag{8}$$

$$H_2O_2 + TiO_2 \ (e_{CB}^-) \rightarrow TiO_2 + HO^\bullet + HO^- \tag{9}$$

*Enhancer: Ag*

$$Ag + TiO_2 \ (e_{CB}^-) \rightarrow Ag^- + TiO_2 \tag{10}$$

$$Ag^- + O_2 \rightarrow Ag + O^{2-} \tag{11}$$

$$O^{2-} + TiO_2 \ (e_{CB}^-) + H^+ \rightarrow TiO_2 + H_2O_2 \tag{12}$$

*Photosensitizer: acetone*

$$CH_3COCH_3 + h\upsilon \rightarrow CH_3CO^\bullet + {}^\bullet CH_3 \tag{13}$$

$$CH_3COCH_3 + h\upsilon \rightarrow CH_3COCH^{\bullet}_2 + H^{\bullet} \qquad (14)$$

$$CH_3COCH^{\bullet}_2 + O_2 \rightarrow CH_3COCH_2OO^{\bullet} \qquad (15)$$

$$CH_3COCH_2OO^{\bullet} + CH_3COCH_3 \rightarrow CH_3COCH_2OOH + CH_3COCH^{\bullet}_2 \qquad (16)$$

$$CH_3COCH_2OOH \rightarrow CH_3COCH_2O^{\bullet} + HO^{\bullet} \qquad (17)$$

$$^{\bullet}OH + MB \rightarrow MB \text{ degradation products} \qquad (18)$$

Under the action of UV radiation, small amounts of acetone led to the formation of OH radicals that increase the rate of the methylene blue degradation reaction.

Considering this reactions network, a scheme concerning the degradation of MB on TiO$_2$-Ag nanoparticles in the presence of acetone under UV irradiation was proposed (Figure 8).

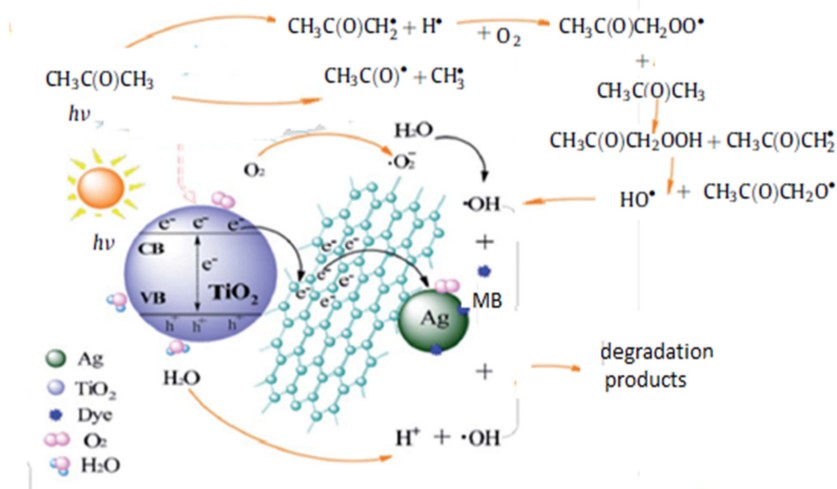

**Figure 8.** Suggested mechanism of MB photodegradation for the TiO$_2$-Ag nanoparticles.

*3.3. Kinetic Analysis*

The photocatalytic oxidation of different dyes is often fitted using the pseudo-first-order and second-order models. Good data fitting between adsorption models and decolorization suggests an outstanding influence of the photocatalyst surface [44–46] on the overall process rate. The mathematical equation of the Langmuir model is given by:

$$-r_{dye} = dC_{dye}/dt = k\ K\ C_{dye}/(1 + K\cdot C_{dye}) \qquad (19)$$

where $r_{dye}$—decomposition dye rate, k—reaction rate constant, K—adsorption equilibrium constant and $C_{dye}$—concentration of dye. At low concentrations values, the denominator of the equation is close to 1, and an equation simplification to a pseudo-first-order Equation (20) is possible:

$$-dC_{dye}/dt = k\cdot K\cdot C_{dye} = k_{obs}\cdot C_{dye} \qquad (20)$$

where $k_{obs}$—apparent pseudo-first-order rate constant, cumulating k and K.

The integration between t = 0 and t, and concentration values between $C_0$ and C, give a simplified linear form of the equation, as:

$$-\ln C_{dye}/C_{dyeo} = k_{obs}\cdot t\cdot(\text{or } -\ln A/A_0 = k_{obs}\cdot t) \qquad (21)$$

The fitting of the decolorization data in time (Equation (21) and the determination of the slopes from the pseudo-first-order kinetic law are shown in Figure 9. The calculated correlation coefficients are presented in Table 3.

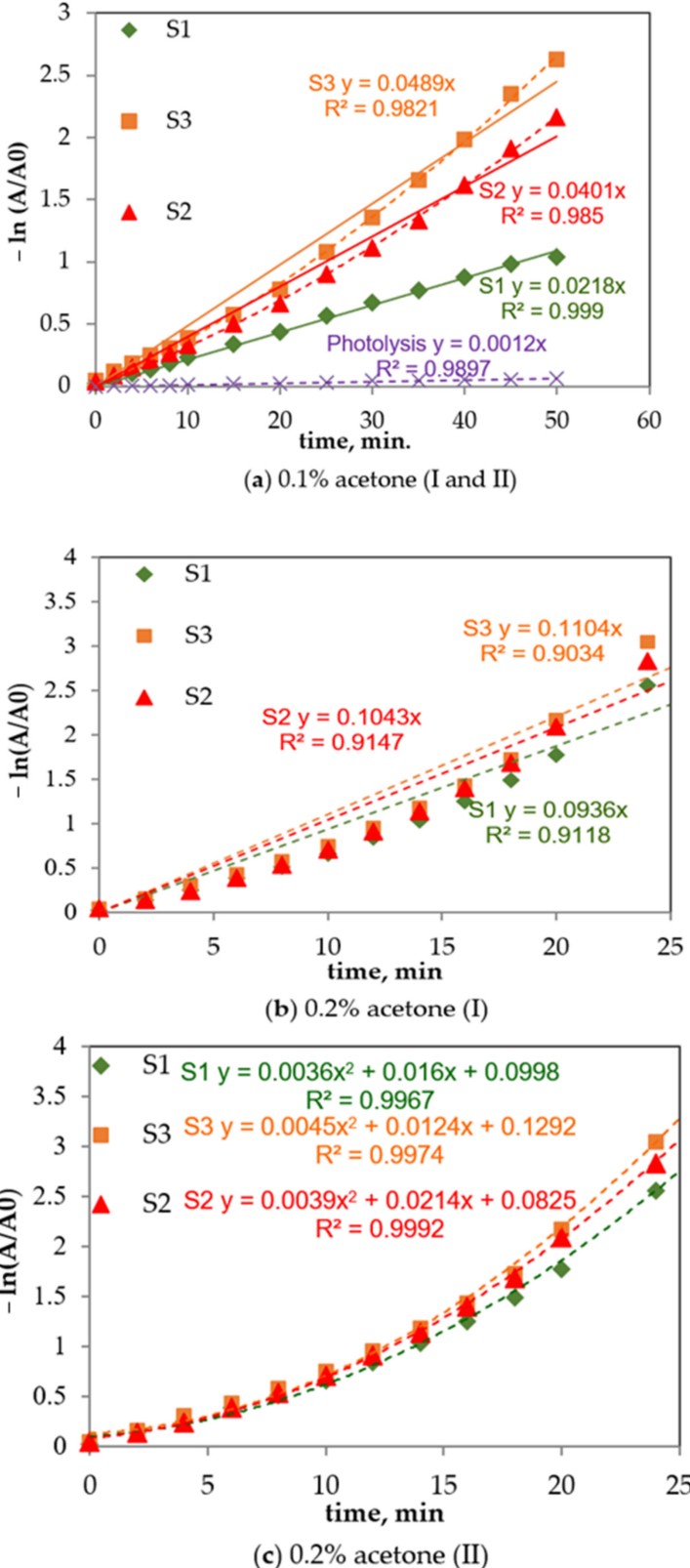

**Figure 9.** (**a**,**b**) Fits of the pseudo-first-order kinetic model for the photocatalytic MB degradation using different sensitizer concentrations and (**c**) second-order model fit for 0.2% acetone. (MB 4 ppm, room temperature, 0.3 g L$^{-1}$ dose).

**Table 3.** $k_{obs}$ values and regression coefficients for describing the photocatalytic kinetics.

| Type of Catalyst | without Acetone | | 0.1% Acetone | | 0.2% Acetone 1st Order | | 0.2% Acetone, 2nd Order |
|---|---|---|---|---|---|---|---|
| | $k^{\circ}_{obs}$ (min$^{-1}$) | $R^2$ | $k_{obs}$ (min$^{-1}$) | $R^2$ | $k_{obs}$ (min$^{-1}$) | $R^2$ | $R^2$ |
| S1 | $9 \times 10^{-4}$ | 0.9974 | 0.0218 | 0.9973 | 0.0936 | 0.9118 | 0.9967 |
| S2 | 0.001 | 0.9939 | 0.0401 | 0.985 | 0.1104 | 0.9147 | 0.9974 |
| S3 | 0.0012 | 0.9926 | 0.0489 | 0.9821 | 0.1043 | 0.9034 | 0.9992 |

The pseudo-first-order model fits well (correlation coefficients over 98%) with decolorization by photolysis and with dye photocatalytic degradation in the presence of 0.1% acetone, on all photocatalysts. The values of the apparent rate constants $k^{\circ}_{obs}$ shown in Table 2 indicate that the reaction rate increases 24 times in the presence of acetone on S1, and more than 40 times on S2 and S3. This is, on one hand, an indication that the presence of acetone modified the photodegradation mechanism, as previously mentioned [47], and that the Ag-doped samples S2 and S3 are more active than S1.

To fit the results for samples S2 and S3, a second-order kinetic model has been shown be better than a pseudo-first-order model (Figure 9a, dotted lines). Indeed, the kinetic equations describing the experimental data are:

$$S2: -\ln (A/A_o) = 0.0004x^2 + 0.0337x, R^2 = 0.9985 \tag{22}$$

$$S3: -\ln (A/Ao) = 0.0003x^2 + 0.0286x, R^2 = 0.9991 \tag{23}$$

The behavior of the systems containing 0.2% photosensitizer when fitting the experimental data to the pseudo-first-order model gave correlation coefficient $R^2$ values around 90%, suggesting that these kinetics are not proper for describing the experimental data. For qualitative purposes, it is however worth examining the values of the $k^{\circ}_{obs}$ values from Table 3, which highlight the very strong increase in the apparent reaction rate for all samples compared with the corresponding values obtained at 0.1% acetone. When the data were fitted to the second-order model, excellent correlation coefficients, above 0.99, were revealed for all three samples.

In order to obtain the influence of acetone content over the apparent rate constant $k_{obs}$, the dependence in Figure 10 can be used. The reaction rate depends strongly on the acetone concentration [48–50].

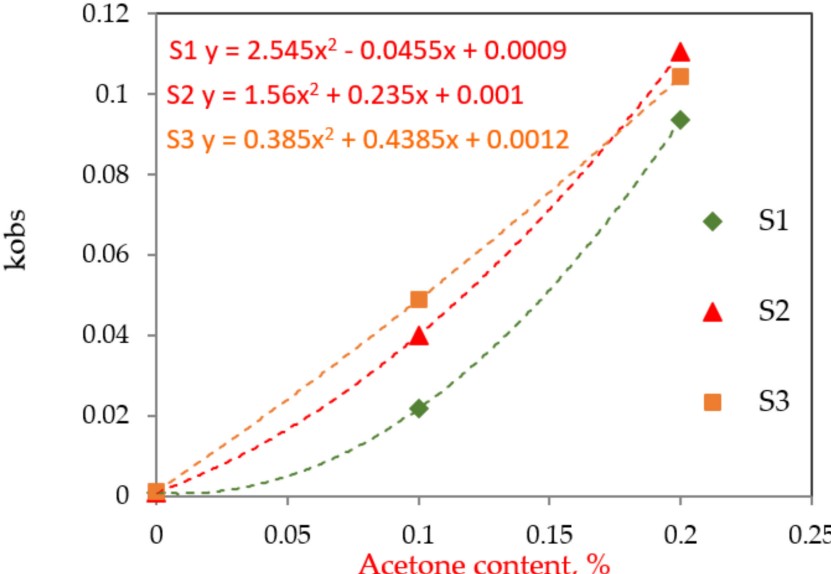

**Figure 10.** Influence of acetone concentration over reaction rate.

The calculated values for MB mineralization after 150 min of reaction were 22.1% for all samples without acetone. On the contrary, the addition in the reaction media of 0.1% and 0.2% acetone enhanced the mineralization efficiency to 54.32% and 64.21%, respectively, after a reaction time of 120 min, which is probably due to the fast conversion of the formed by-products (data not shown).

The degradation of MB was subjected to several photocatalytic cycles with the same catalyst in order to assess its reusability and stability in the reaction. According to Figure S1 (in Supplementary Materials), good catalyst recyclability was found in terms of pollutant removal efficiency, with only 5% activity loss after four successive cycles (details of the reusability are shown in the supporting information).

## 4. Conclusions

In this study, $TiO_2$-Ag nanocomposites with superior photocatalytic performance have been synthesized by the sol-gel method, using titanium tetraisopropoxide and silver nitrate as precursors. The SEM and TEM investigations highlight the presence of silver nanoparticles. The calcination modified the initial morphology of the $TiO_2$ sample, favoring the uniform dispersion of Ag on the surface better than on the as-synthesized sample. After calcination at 650 °C, a rutile phase appeared next to the initial anatase, and some of the Ag nanoparticles were grouped in big cluster and detected by XRD. The incorporation of Ag effectively improved the photocatalytic activity of $TiO_2$ by controlling the carrier charge recombination.

The materials showed good efficiency in the degradation of methylene blue dye, especially on the Ag-doped samples and when acetone was employed as the photosensitizer. At a 0.1% acetone ratio in the reaction mixture, over 90% of the MB was decomposed in 50 min, and with 0.2% acetone added in the reaction mass, the time for reaching the same decolorization yield decreased by 25 min. A graphical construction was conceived for explaining the MB degradation mechanism, and the kinetic models of pseudo-first- and pseudo-second-order models were successful used to fit the experimental data.

After 120 min of reaction, mineralization yields of 54.32% and 64.21% were achieved in the presence of 0.1% acetone and 0.2% acetone, respectively.

Moreover, it was found that the $TiO_2$-Ag sample displays an efficient recyclability with only 5% activity loss after four consecutive photocatalytic cycles, which is very promising for its use in future practical applications.

In summary, a financially accessible and environmentally friendly approach has been proposed for MB degradation, involving small concentrations of $TiO_2$-Ag nanostructured catalysts and low acetone content, with a view to improving the degradation reaction and its mineralization efficiency. The proposed methodology is very promising from an economic point of view, minimizing the treatment costs due to the reduction of the catalyst dose and the energy consumption during the irradiation step. Additionally, the improvement observed in the mineralization yield is very beneficial for the reduction in the deleterious effects of the treated solutions.

**Supplementary Materials:** The following supporting information can be downloaded at: https://www.mdpi.com/article/10.3390/w14172711/s1, Figure S1: Variation of MB photodegradation efficiency within four consecutive reaction cycles.

**Author Contributions:** Conceptualization, M.H., D.L. and G.A.A.; methodology, C.N.D., L.F. and G.A.A.; validation, G.C. and L.F.; formal analysis, G.A.A. and C.G.d.C.; investigation, C.G.d.C. and C.N.D.; resources, M.H.; writing—original draft preparation, G.C. and C.N.D.; writing—review and editing, D.L. and M.H.; visualization, C.G.d.C.; supervision, M.H. and D.L. All authors have read and agreed to the published version of the manuscript.

**Funding:** This research received no external funding.

**Institutional Review Board Statement:** Not applicable.

**Informed Consent Statement:** Not applicable.

**Data Availability Statement:** The data presented in this study are available on request from the corresponding author.

**Conflicts of Interest:** The authors declare no conflict of interest.

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
