# Peer review of "Enhancing the TiO2-Ag Photocatalytic Efficiency by Acetone in the Dye Removal from Wastewater"

_water, doi:10.3390/w14172711_

Round 1

Reviewer 1 Report

TiO2 nanoparticles synthesized by sol-gel method and doped with Ag, were characterized by SEM, EDAX, FTIR, XRD and TEM, then tested in the photocatalytic degradation of methylene blue (MB), under UV irradiation. The experimental results indicated that the average size of the raw particles was 10 nm and their size increased by calcination. The photocatalytic degradation of MB on nanostructured TiO2-Ag shows a high degradation efficiency upon the addition of a photosensi-tizer. A MB degradation mechanism was proposed to explain the synergy between the TiO2 and the silver nanoparticles in the degradation performance. This simple approach could be applied for the advanced cleaning of wastewater contaminated with dyes, in the perspective of its reuse. Therefore, this work is worth publishing, but this manuscript needs minor revision. My detailed comments are as follows:

1. TiO2-Ag photocatalyst has been applied in other fields and should be compared with it.

2. In the preparation of the experiment, a space should be left between 110℃.

3. In XRD, S1 samples should also be displayed for comparison.

4. In the SEM diagram, TiO2 and Ag should be marked to distinguish; In addition, HRTEM images should be supplemented to observe its morphology.

5. The stability of the prepared photocatalyst should be explained by supplementary experiments. Some similar works can be recommended and cited. The author can refer to these papers, Journal of Hazardous Materials 410 (2021) 124539. Chemical Engineering Journal 409 (2021) 128185. Journal of Cleaner Production 360 (2022) 131948.

6. In Figure 7, when the proportion of acetone is 0.2%, the author did not explain why the decolorization rate of MB of the three samples is the same.

Author Response

Response to comments of Reviewer 1 and list of changes made in the manuscript:

" Enhancing the TiO2-Ag photocatalytic efficiency by acetone in the dye removal from wastewater"

We would like to thank the reviewer for comments and suggestions for improving the scientific quality of our manuscript. We have carefully considered all the reviewer comments and have revised the manuscript in light of them. The suggested modifications were clearly marked in red in the revised manuscript. Details of our responses to reviewer comments are shown below. We hope you will find these revisions rise to your expectations

Reviewer 1

TiO2 nanoparticles synthesized by sol-gel method and doped with Ag, were characterized by SEM, EDAX, FTIR, XRD and TEM, then tested in the photocatalytic degradation of methylene blue (MB), under UV irradiation. The experimental results indicated that the average size of the raw particles was 10 nm and their size increased by calcination. The photocatalytic degradation of MB on nanostructured TiO2-Ag shows a high degradation efficiency upon the addition of a photosensitizer. A MB degradation mechanism was proposed to explain the synergy between the TiO2 and the silver nanoparticles in the degradation performance. This simple approach could be applied for the advanced cleaning of wastewater contaminated with dyes, in the perspective of its reuse. Therefore, this work is worth publishing, but this manuscript needs minor revision. My detailed comments are as follows:

  1. TiO2-Ag photocatalyst has been applied in other fields and should be compared with it.

We thank the Reviewer 1 for this valuable suggestion. According to Reviewer comment new information related to the application in other fields of TiO2-Ag system and new references were added in the Introduction section of the revised manuscript (lines 78-80 and references 27 and 28).

  1. In the preparation of the experiment, a space should be left between 110℃.

We thank the Reviewer 1 for this useful remark. This correction was done in the revised manuscript.

  1. In XRD, S1 samples should also be displayed for comparison.

We thank again the Reviewer 1 for this important and useful remark. Accordingly Figure 1 was completed with the missing information.

  1. In the SEM diagram, TiO2and Ag should be marked to distinguish; In addition, HRTEM images should be supplemented to observe its morphology.

We thank the Reviewer 1 for this interesting suggestion. There is no possibility, unfortunately, to distinguish in our case individual morphologies for the Ag particles deposited on TiO2. Therefore, the TEM measurements were performed. There the dark dots were identified by the SAED images. In order to clarify some of these aspects new information was included in the revised manuscript as follows:

“The fine dispersion of the Ag nanoparticles on titania after the calcination was highlighted by TEM analysis (Figure 3). The Ag nanoparticles are present as small crystalline grains (dark dots) seldom agglomerated, with sizes less than 20 nm”.

We hope that the revised version will meet your expectations.

5.-The stability of the prepared photocatalyst should be explained by supplementary experiments. Some similar works can be recommended and cited. The author can refer to these papers, Journal of Hazardous Materials 410 (2021) 124539. Chemical Engineering Journal 409 (2021) 128185. Journal of Cleaner Production 360 (2022) 131948.

We thank once again the Reviewer 1 for this important and useful remark. A new paragraph and a new figure highlighting the catalyst stability were included in the Section 3.3 (lines 344-347) and in the “Supplementary information section” of the revised manuscript. Moreover, the interesting papers suggested by the reviewer were also considered in the revised form of our manuscript (references 24-26).

  1. In Figure 7, when the proportion of acetone is 0.2%, the author did not explain why the decolorization rate of MB of the three samples is the same.

We thank the Reviewer 1 for this recommendation. According to your suggestion the revised manuscript was completed follows:

…. acetone can split in radicals upon UV irradiation, generating extra OH• radicals in the reaction medium and enhancing the photoactivity. More radical particles are present in the reaction medium, more dye molecules will be fragmented, enhancing thus the degradation of the target compound (the decolorization was close in value 95.2, 96.42 and 97.25%, respectively).

Reviewer 2 Report

The paper focus on the oxidation of dyes usingTiO2-Ag nanocomposite enhanced by adding acetone. The author also investigated the catalytic characterizations with kinetics modeling

 In general, the results mostly support the authors' conclusions. However, some aspects of the manuscript must be carefully reviewed, discussed and improved.

1°) The originality, mechanism, and scientific reliability of the work are unclear. In my opinion, there are some major points that the authors should address before it is accepted for publication.

2°) The abstract is not informative and does not give major quantitative results like the kinetics. 

3°) Why do authors study the removal of these compounds: dyes? Authors must indicate why the removal of these compounds could be interesting. 

4°) please give the specific surface of catalysts (what about the adsorption part)

5°) What about the reaction mechanism? Authors are invited to consolidate the scientific part of paper. Please add more data on actives species ( OH°, O2- ?).  In order to be more correct, I suggest adding more references to manuscript (CHERD 144, 300-309 (2019): about photocatalytic treatment: Materials 12 (3), 412 (2019) : ROS

6°) What about the mineralization rate of dyes  

7°) I am not agree with the modelling of experimental kinetics with the pseudo-first-order (see figure 9 ). Explanation are needed here.

8°) Please add the errors bars in order to avoid confusion

9°) what about the reusability of catalyst TiO2-Ag nanocomposite.

Author Response

Response to comments of Reviewer 2 and list of changes made in the manuscript:

" Enhancing the TiO2-Ag photocatalytic efficiency by acetone in the dye removal from wastewater"

We would like to thank the reviewer for the valuable comments and suggestions for improving the scientific quality of our manuscript. We have carefully considered all the reviewer comments and have revised the manuscript in light of them. The suggested modifications were clearly marked in red in the revised manuscript. Details of our responses to reviewer comments are shown below. We hope you will find these revisions rise to your expectations.

Reviewer 2

The paper focus on the oxidation of dyes usingTiO2-Ag nanocomposite enhanced by adding acetone. The author also investigated the catalytic characterizations with kinetics modeling

 In general, the results mostly support the authors' conclusions. However, some aspects of the manuscript must be carefully reviewed, discussed and improved.

1°) The originality, mechanism, and scientific reliability of the work are unclear. In my opinion, there are some major points that the authors should address before it is accepted for publication.

We thank the Reviewer 2 for all recommendations that made possible to improve the scientific quality of our manuscript.

2°) The abstract is not informative and does not give major quantitative results like the kinetics. 

We thank again the Reviewer 2 for this important and useful remark. According to the reviewer’s suggestions in the revised manuscript the abstract was completed in order to be more informative and to give quantitative results. Please see the lines 25-32.

3°) Why do authors study the removal of these compounds: dyes? Authors must indicate why the removal of these compounds could be interesting. 

We thank the Reviewer 2 for this valuable remark. According to the reviewer suggestion the missing information were considered in the Introduction section of the revised manuscript.

4°) please give the specific surface of catalysts (what about the adsorption part).

We thank Reviewer 2 for this recommendation. The missing information was provided in the revised manuscript. The measurement performed with the Nova 2200 apparatus allowed the determination of the specific surface area using the BET equation. The calculated values of the surface area of the samples were respectively 20.35 m2/g for the S1, 25.37 m2/g for the S2 and 21.69 m2/g for S3. Hope that this new version fits with your expectations.

5°) What about the reaction mechanism? Authors are invited to consolidate the scientific part of paper. Please add more data on actives species (OH°, O2- ?).  In order to be more correct, I suggest adding more references to manuscript (CHERD 144, 300-309 (2019): about photocatalytic treatment: Materials 12 (3), 412 (2019) : ROS

We thank Reviewer 2 for these useful recommendations. This part of our manuscript was completed according with the recommended references and the suggested references were included in the revised form of our manuscript (as references 14 and 43).

6°) What about the mineralization rate of dyes  

We thank again Reviewer 2 for this important and useful remark. The mineralization of MB was also measured and new data related to pollutant mineralization were provided in the revised manuscript (Materials and methods and Results sections).

“The mineralization of MB was expressed as the percentage of total organic carbon (TOC) reduction, calculated from the difference between the initial and final TOC content measured for the aqueous dye solution:

                                 (2)

For the total organic carbon measurement, a TOC-L Shimadzu equipment (TOC-L Shimadzu, Milano, Italy) was used”.

“The calculated values for MB mineralization after 150 minutes of reaction were 22.1% for all samples without acetone. On the contrary, the addition in the reaction media of 0.1% and 0.2% acetone enhanced the mineralization efficiency to 54.32% and 64.21%, respectively after a reaction time of 120 minutes, which is probably due to the fast conversion of the formed by-products”.

7°) I am not agree with the modelling of experimental kinetics with the pseudo-first-order (see figure 9). Explanation are needed here.

We thank again Reviewer 2 for this important and useful remark. The discussion related to the modeling of experimental kinetics was completed to enhance the understanding of the suggested aspects as indicated below:

“The values of the apparent rate constants koobs shown in table 2 indicate that the reaction rate increases 24 times in the presence of acetone on S1 and more than 40 times on S2 and S3. This is on one part an indication that the presence of acetone modified the photodegradation mechanism, as previously mentioned [32], and that the Ag-doped samples S2 and S3 are more active than S1.

An attempt to fit the results for samples S2 and S3 to a second degree kinetic mod-el revealed a better fit than in the case of pseudo-first degree (figure 9 a, dotted lines). Indeed, the kinetic equations described the experimental data were, respectively:

S2: - ln (A/Ao) = 0.0004 x2 + 0.0337x, R2 = 0.9985                                                            (6)

S3: - ln (A/Ao) = 0.0003x2 + 0.0286x, R2 = 0.9991                                                 (7)

The behavior of the systems containing 0.2% photosensitizer when fitting the ex-perimental data to the pseudo-first order model gave correlation coefficients R2 have values around 90%, suggesting that this kinetics it is not proper for describing the ex-perimental data. For qualitative purposes, it is however worth examining the values of the koobs values from Table 2, which highlight the very strong increase of the apparent reaction rate for all samples compared with the corresponding values obtained at 0.1% acetone. When the data were fitted to the second degree model, excellent correlation coefficients, above 0.99, were revealed for all three samples.”

Hope that the revised version of our manuscript will fit your expectations.

8°) Please add the errors bars in order to avoid confusion

We thank Reviewer 2 for this recommendation. The error bars were added to avoid confusion.

9°) what about the reusability of catalyst TiO2-Ag nanocomposite.

We thank again Reviewer 2 for this important and useful remark. These aspects were also provided in the revised form of our manuscript (results section and Supplementary material).

“The degradation of MB was subjected to several photocatalytic cycles with the same catalyst in order to assess its reusability and stability in reaction. According to Figure S1, a good catalyst recyclability was found in terms of pollutant removal efficiency with only 5% of activity loss after four successive cycles (details of the reusability are in the supporting information)”.

We hope you will find these corrections rise to your expectations.

Reviewer 3 Report

Manuscript Number: water-1869170

In this study, TiO2-Ag nanocomposites with superior photocatalytic performance have been synthesized by sol–gel method from titanium tetraisopropoxide and silver nitrate as precursors. The manuscript is suitable for publication only after major revision.

1.     The author should use tables to compare the removal effects of different materials on MB.

2.     All degradation experiments should be repeated three times and use error sticks in the figure.

3.     The author should give the reasons for the practical use of this work from three aspects: cost, operability and accuracy.

4.     It is suggested that the author test the actual industrial wastewater containing dyes.

5.     Heavy metal ions are a kind of pollutants that harm the environment. Several relative papers are suggested to be cited. (10.1021/acsami.1c22035, 10.1016/j.clay.2018.12.017).

6.     How does the author prove that the methylene blue dye is a degradation phenomenon rather than an adsorption behavior?

Author Response

Response to comments of Reviewer 3 and list of changes made in the manuscript:

" Enhancing the TiO2-Ag photocatalytic efficiency by acetone in the dye removal from wastewater"

We would like to thank the reviewer for comments and suggestions for improving the scientific quality of our manuscript. We have carefully considered all the reviewer comments and have revised the manuscript in light of them. The suggested modifications were clearly marked in red in the revised manuscript. Details of our responses to reviewer comments are shown below. We hope you will find these revisions rise to your expectations

Reviewer 3

In this study, TiO2-Ag nanocomposites with superior photocatalytic performance have been synthesized by sol–gel method from titanium tetraisopropoxide and silver nitrate as precursors. The manuscript is suitable for publication only after major revision.

  1. The author should use tables to compare the removal effects of different materials on MB.

We thank Reviewer 3 for this useful recommendation. Accordingly, a new Table and 6 new references were included in the revised manuscript to complete the missing information. Please see Table 2 and the reference list.

We hope you will find these corrections rise to your expectations.

  1. All degradation experiments should be repeated three times and use error sticks in the figure.

We thank Reviewer 3 for this recommendation. Indeed, all the experiments were conducted in triplicate. The missing information was provided in the “Materials and methods” section (Page 3, line 136) as well as the error bars (please see figure 6).

  1. The author should give the reasons for the practical use of this work from three aspects: cost, operability and accuracy.

We thank Reviewer 3 for this important and useful suggestion. Supplementary comments were included in the revised manuscript to support the suggested aspects. Please see lines 374-381. We hope you will find these corrections rise to your expectations.

  1. It is suggested that the author test the actual industrial wastewater containing dyes.

We thank again the Reviewer 3 for this important and useful remark.

This paper aimed the preparation of a simple photocatalyst, better that the genuine TiO2, by Ag doping. Since methylene blue is a very stable molecule, many papers dealing with photocatalysis use it as a model molecule. In this step of investigation, we targeted just to prove that the association between Ag-doped TiO2 and use of extremely low acetone doses brings a very fast and efficient dye removal. The colored compounds removal from industrial wastewater are of course interesting, but were not in our attention when we planned this work. However, for our future studies we will take into account the valuable remark of Reviewer 3 in order the check the efficiency of the proposed process under conditions near to real ones.

  1. Heavy metal ions are a kind of pollutants that harm the environment. Several relative papers are suggested to be cited. (10.1021/acsami.1c22035, 10.1016/j.clay.2018.12.017).

We thank again the Reviewer 3 for this important and useful remark. We completed the manuscript according to your suggestion and as well as with the missing references:

“The contaminants found in wastewater, including heavy metal ions, organic compounds, microorganisms, etc. have raised serious concerns owing to their significant negative effects on the environment and the economy [1-4]”.

  1. How does the author prove that the methylene blue dye is a degradation phenomenon rather than an adsorption behavior?

We thank the Reviewer 3 for this comment.

The collected data of the adsorption experiments conducted in dark clearly showed that the adsorption equilibrium was achieved after a contact time of 25 minutes. After this period the measured absorbance remained quite constant. The pollutant degradation was confirmed by the decolorization of the treated solutions as well as by the results of the TOC measurements.

Round 2

Reviewer 3 Report

Accept in present form